# Factors influencing the utilisation of anti-HBs titre testing services among nursing students in Northwest Ghana: A cross-sectional study

**Augustine Ngmenemandel Balegha** *

Department of Obstetrics and Gynaecology, St. Theresa's Hospital, Nandom, Upper West Region, Ghana, West Africa

* bnaugustine@gmail.com

## Abstract

Understanding the impact of hepatitis B testing, vaccination, the number of vaccine doses, and socio-demographics on post-vaccination anti-HBs titre testing, is essential for hepatitis B prevention. The aim of the study was to determine the prevalence of hepatitis B testing, hepatitis B vaccination, number of vaccine doses received, socio-demographic characteristics, and their impact on anti-HBs titre testing among nursing students in Northwest Ghana. A stratified sample of 402 nursing students from Wa and Lawra nursing colleges in Ghana's Upper West Region was surveyed in November 2020 using an online cross-sectional design. STATA 13 was used to analyse the data, which described socio-demographics, hepatitis B testing, hepatitis B vaccination, and post-vaccination anti-HBs titre testing with frequencies and percentages. Hierarchical binary logistic regression models were used to investigate the relationships between post-vaccination anti-HBs titre testing and hepatitis B testing, vaccination, the number of vaccine doses, and their socio-demographics. The study discovered that while hepatitis B testing was high (89.0%), rates for hepatitis B vaccination (72.1%), obtaining the recommended vaccine doses (59.5%), and post-vaccination anti-HBs titre testing (19.4%) were lower. Nursing students who accepted hepatitis B vaccination were significantly more likely to undergo anti-HBs titre testing [aOR = 12.34; 95% CI = 1.80–84.54; $p < 0.05$]. Those who received $\geq 3$ vaccine doses were over 8 times more likely to utilise anti-HBs titre testing [aOR = 8.31; 95% CI = 2.73–25.34; $p < 0.001$]. Wa NTC students were 74% less likely to access anti-HBs titre testing [aOR = 0.26; 95% CI = 0.15–0.47; $p < 0.001$]. Students with parents who had tertiary education were significantly more likely to undergo anti-HBs titre testing [aOR = 2.50; 95% CI = 1.42–4.42; $p < 0.01$]. The study reveals high hepatitis B testing but low vaccination rates, emphasizing the need for required vaccine doses and post-vaccination anti-HBs testing. Key predictors include hepatitis B vaccination, $\geq 3$ doses, Wa NTC enrollment, and parental education. The study advocates mandatory testing, vaccination, and affordable access to anti-HBs titre testing. Unvaccinated students, those with <3 doses, Wa NTC attendees, and those with lower parental education for nursing school admission should be prioritised.

**Data Availability Statement:** The minimal dataset for this study are contained in the supporting information file.

**Funding:** The authors received no specific funding for this work.

**Competing interests:** The authors have declared that no competing interests exist.

## Introduction

Hepatitis B infection, caused by the hepatitis B virus, is a global public health threat [1]. Global annual hepatitis B morbidity and mortality remain unacceptably high [2]. Sub-Saharan Africa and Southeast Asia have the highest hepatitis B morbidity and mortality rates [2]. Hepatitis B infection is still highly prevalent in Ghana, at 12.3% [3]. According to the Advisory Committee for Immunisation Practices of the Centres for Disease Control and Prevention, effective hepatitis B infection prevention necessitates hepatitis B testing, complete hepatitis B vaccination, and post-vaccination anti-HBs titre testing [4–6]. Hepatitis B testing detects the presence of the hepatitis B surface antigen (HBsAg) or viral DNA [7]. Effective protection against hepatitis B infection requires complete vaccination (at least three doses of the hepatitis B vaccine at 0-, 1-, and 6-month intervals) and anti-HBs titre testing 1–2 months after completion of the vaccine series [8]. Anti-HBs titre testing determines the level of immunity (anti-HBs titre > 10 IU/ml is recommended) conferred by the hepatitis B vaccine post-vaccination. Post-vaccination anti-HBs titre < 10 IU/ml is a requirement for booster doses of the vaccine [8]. In line with Sustainable Development Goal 3.3 of eliminating viral hepatitis by 2030, the importance of hepatitis B testing, complete hepatitis B vaccination, and post-vaccination anti-HBs titre testing has been highlighted by the Advisory Committee for Immunisation Practices and the Global Health Sector Strategy on Viral Hepatitis 2016–2021 [6, 9].

However, previous studies have revealed suboptimal rates of hepatitis B testing, complete vaccination, and post-vaccination anti-HBs titre testing. The hepatitis B testing rate was reported as 16.4% among medical students in Syria [10], 70% among nurses in Nigeria [11], and 49.6% among undergraduate health science students in Ghana [4]. Complete hepatitis B vaccination rate was reported as 60% among healthcare workers in China [12], 18% among medical students in Cameroon [13], and 30.5% among health science students in Ghana [4]. Low anti-HBs titre testing rates of 7% have been reported among nursing students in India [14], 29% among doctors and nurses in Nigeria [15], and 19.4% among nursing students in Ghana [1]. Low prevalence rates of hepatitis B infection testing, complete hepatitis B vaccination, and post-vaccination anti-HBs titre testing is linked to a higher incidence of hepatic cirrhosis and hepatocellular carcinoma, which are significant contributors to global hepatitis B infection-specific mortality [16]. Also, based on empirical evidence, hepatitis B testing, complete hepatitis B vaccination, number of hepatitis B vaccine doses received and socio-demographic characteristics (sex, age, marital status, ethnicity, religion, residential environment, college, programme of study, level of study, and parents' education) can predict post-vaccination anti-HBs titre testing [4, 5, 12, 15, 17].

While studies on the prevalence of hepatitis B testing, complete vaccination, and post-vaccination anti-HBs titre testing and their socio-demographic correlates exist, most of these studies have been conducted among healthcare workers and not their trainees, who carry a greater risk of infection [18]. Besides, there is limited and fragmented empirical evidence on the predictors of post-vaccination anti-HBs titre testing among the poorly researched high-risk nursing students. There is also a dearth of studies exploring the link between hepatitis B testing, vaccination, the number of doses of vaccine taken, and post-vaccination anti-HBs titre testing. Therefore, the objectives of this study were to assess the influence of hepatitis B testing, hepatitis B vaccination, the number of vaccine doses taken and relevant socio-demographics, on post-vaccination anti-HBs titre testing among nursing students in the resource-poor Northwest Ghana. The findings of this study can help policymakers create a comprehensive strategy for effective hepatitis B infection prevention and control.

## Methods

### Theoretical framework

Several theories have been proposed to explain health-seeking behaviours, including healthcare access and utilisation. This study is positioned within the framework of Andersen's healthcare utilisation model [19] and Levesque et al. [20]'s healthcare access model. Andersen's healthcare utilisation model, originally proposed in 1968, posits that, utilisation of healthcare services is dependent on three-pronged dynamics of predisposing factors (race, age, health attitudes), enabling factors (family support, access to health insurance), and need factors (perceived and actual). Levesque et al. [20] defined access to healthcare as the ability to identify healthcare needs, seek healthcare services, reach, obtain, or use health care services, and actually have a need for services met. Levesque et al. [20] based their argument on five criteria of access: approachability, acceptability, availability, accommodation, cost, and appropriateness.

In the context of the Andersen [19] healthcare utilisation model and the Levesque et al. [20] healthcare access dimensions model, this study hypothesises that nursing students from various socio-demographic backgrounds, including health beliefs, would access or utilise the needed healthcare service in the presence of enabling factors such as family support and access to health insurance and have a need for access to healthcare.

### Study setting and population

The study was conducted in the Upper West Region (UWR) of Ghana. The UWR shares borders with Burkina Faso to the north and west, Northern Region to the south and Upper East Region to the east [21]. The region lies between latitudes 9˚30'N and 11˚N and longitudes 1˚ 25'W and 2˚45'W [21]. Currently, the region has 1 regional hospital, 4 district hospitals, 2 Christian Health Association of Ghana (CHAG) hospitals, 3 private hospitals, 4 polyclinics, 72 health centres, 10 clinics, 5 maternity homes, and 319 Community Health Planning Services (CHPS) compounds [22]. The doctor-to-patient ratio in the region is estimated to be 1:16,222 (national figure of 1:8,098), while the nurse-to-patient ratio is 1:291 (national average of 1:478) [23]. Furthermore, 62% of the residents of the region travel at least 30 minutes to access healthcare [24]. The UWR was chosen because it is one of the poorest regions with little research on hepatitis B infection prevention, including its prevalence [25]. To the best of my knowledge, no research linking hepatitis B testing, hepatitis B vaccination, number of vaccine doses and post-vaccination anti-HBs titre testing has been conducted in the region. As a result, the findings of this study can provide effective hepatitis B prevention context-specific policy direction.

This research focused on nursing students at the Wa and Lawra NTCs, which are located in the Wa and Lawra municipalities, respectively [21]. Wa Municipality is bounded by Nadowli-Kaleo District to the North, Wa East District to the east and west and Wa West Dstrict to the south [21]. Wa Municipality lies within latitudes 1˚40'N to 2˚45'N and longitudes 9˚32'W to 10˚20'W [21]. Lawra Municipality on the other hand shares boundaries with Nandom Municipal to the north, east by Lambussie-Karni District, south and west by Burkina Faso [21]. Lawra District lies within latitudes 10˚20'N and 11˚00'N and longitudes 2˚25'W and 2˚45'W [21]. Nursing students were purposefully chosen because they are at higher risk of infection when compared to healthcare workers and yet remain poorly researched [18]. There are currently seven NTCs located in four of the region's eleven districts. Wa and Lawra NTCs were chosen purposively for their context (urban versus rural) and study programmes (Registered General Nursing [RGN] versus Nurse Assistant Curative [NAC]). At the time of data collection, Wa and Lawra NTCs had 399 and 303 students, respectively. However, students who were in their last year of college were not available to participate in the study.

**Table 1. Probability proportional to stratal size sampling strategy.**

| Level of study | Population per class | Percentage (%) | Proportional sample size |
|---|---|---|---|
| Registered General Nursing 1 (Wa) | 58 | 13.5 | 55 |
| Registered General Nursing 2 (Wa) | 68 | 15.8 | 65 |
| Nurse Assistant Curative 1 (Wa) | 114 | 26.5 | 110 |
| Registered General Nursing 1 (Lawra) | 36 | 8.4 | 34 |
| Registered General Nursing 2 (Lawra) | 56 | 13.0 | 53 |
| Nurse Assistant Curative 1 (Lawra) | 98 | 22.8 | 93 |
| **Total** | **430** | **100** | **410** |

1- First year; 2-Second year

## Study design and sampling

A quantitative analytical cross-sectional design was implemented in this study. The sample size for the study was calculated using Cochran [26]'s formula:

$$n = \frac{z^2 pq}{d^2}$$

Where n = minimum estimated sample size; z = 1.96, p = estimated proportion of hepatitis B vaccination coverage among the nursing students; q = 1-p and d = margin of error of 5%. Cochran's formula was used since the total nursing student population in the Upper West Region and Ghana at large was unknown, as such data are not readily available. Therefore, using an assumed 66.8% hepatitis B vaccination coverage [27], 5% error margin, the sample size was calculated as;

$$n = \frac{(1 \cdot 96)^2 \times 0.668(1 - 0 \cdot 668)}{(0.05)^2}$$

$$n = 341$$

To account for non-responses, the inherent low response rate associated with e-surveys [28] and guided by Andrade [29], the sample size was increased by 20% to **410**.

Multi-stage stratified random sampling technique was used to select the study participants. The study population was stratified by college, programme, and level of study. Using an index number-based class register, the lottery system, and a probability proportional to stratal size approach, a simple random sampling of students in each stratum was conducted. Table 1 shows how the probability proportional to stratal size approach was applied.

## Data collection

Data was collected using a self-administered questionnaire. The questionnaire was drawn from a previous study [1]. The original questionnaire was pre-validated (average inter-item correlation coefficient = 0.51) and reliable (overall Cronbach's alpha = 0.71). The details of the study instrument and its pre-testing, validity, and reliability assessments are described elsewhere [1]. However, in this current study, socio-demographic characteristics of the respondents, hepatitis B testing, hepatitis B vaccination, number of doses of hepatitis B vaccine taken, and post-vaccination anti-HBs titre testing among the nursing students were analysed. The socio-demographic characteristics assessed include sex, age, marital status, ethnicity, religion, permanent residence, college, programme of study, level of study, and parents' level of education. Hepatitis B testing, hepatitis B vaccination, and post-vaccination anti-HBs titre testing

were assessed by asking the respondents whether they had been screened for hepatitis B, vaccinated against hepatitis B, and done the post-vaccination anti-HBs titre testing, respectively. The respondents were required to provide a "yes" or "no" response in each case. However, to indicate the number of doses of the hepatitis B vaccine received, respondents had to select either 1, 2, 3, or 4.

Due to the COVID-19 pandemic, the survey was conducted largely online from 6[th] November, 2020 to 27[th] November, 2020. The questionnaire was designed using Google Forms. Each of the study participants was sent the questionnaire via WhatsApp. Paper copies of the questionnaire were supplied to participants who could not complete it online because of challenges with internet connectivity. Non-responders were reminded regularly to complete the questionnaire [28].

## Variables of the study

The study's outcome variable was anti-HBs titre testing. The key explanatory variables were hepatitis B testing, hepatitis B vaccination, and the number of vaccine doses (Table 2). The other explanatory variables were the respondents' socio-demographic characteristics (Table 2). Table 2 presents a description of the variables in the study.

## Data processing and analysis

The responses were retrieved, cleaned, entered, and coded using STATA 13. Only data (S1 Data) from complete cases were included in the analysis.

Frequencies and percentages were used to describe the respondents' socio-demographic characteristics, hepatitis B testing, hepatitis B vaccination, number of vaccine doses, and post-vaccination anti-HBs titre testing. The socio-demographic characteristics of the respondents were examined for multi-collinearity using the variance inflation factor (VIF). There was no collinearity (VIF < 10) among the socio-demographic factors (minimum VIF = 1.03; maximum VIF = 1.79; and mean VIF = 1.25). At the bivariate level, chi square test was performed to determine the association between the outcome variable (post-vaccination anti-HBs titre testing) and the socio-demographic characteristics of the respondents. Factors that were significant at the bivariate level of analysis were then included in a hierarchical multivariable logistic regression analysis. Standard logistic regression models were run. However, due to the presence of sparse data for some cells of the variables and the rareness of the outcome variable (Anti-HBs titre testing), biased estimates were obtained. Large odds ratios with widened confidence intervals were obtained, particularly for hepatitis B vaccination and number of vaccine doses received. Therefore, guided by Heinze and Fuhr [30] and Firth [31], the firthlogit binary logistic regression analyses were performed. The firthlogit analysis produces nearly unbiased effect size estimates that is representative of the population by maximization of the penalized log likelihood method. Four models were fitted. Model 0 was an empty model (containing no predictors). Model 1 contained the outcome variable and the key explanatory variables. Model 2 contained the outcome variable and socio-demographic characteristics. Model 3 contained the outcome variable, the key explanatory variables, and the socio-demographic characteristics of the respondents. The fixed effects results were reported as adjusted odds ratios (aORs) and considered significant at a 5% margin of error and 95% confidence interval. The *firthfit function* was used for model fitness comparison, using penalized log-likelihood, McFadden $R^2$, Akaike's information criterion (AIC) and Bayesian information criterion (BIC). The results were written in line with the Strengthening the Reporting of Observational Studies in Epidemiology (STROBE) statement [28] and the Checklist for Reporting Results of Internet E-surveys (CHERRIES) [32].

**Table 2. Definition/ measurement of study variables.**

| Variable | Definition | Measurement | Reference |
|---|---|---|---|
| **Dependent variables** | | | |
| Hepatitis B testing | Tested for hepatitis B infection or not | 0. No 1. Yes | No |
| Full hepatitis B vaccination | Taken three/four doses or not | 0.Less than three doses 1. Three/four doses | Less than three doses |
| Post-vaccination anti-HBs titre testing | Done post-vaccination anti-HBs titre testing or not | 0. No 1. Yes | No |
| **Independent variables** | | | |
| Sex | Male or female | 0. Female 1. Male | Female |
| Age | Age at last birthday | 0. < 23 years 1. $\geq$ 23 years | < 23 years |
| Marital status | Marital status or never married | 0. Married 1. Never married | Married |
| Religion | Religious affiliation of the respondent | 0. Islam 1. Christianity | Islam |
| Ethnicity | Affiliated ethnic group | 1. Dagaaba/Waala/Sissala 2. Akan 3. Others | Dagaaba/Waala/Sissala |
| Permanent residence | Urban or rural residence | 0. Rural 1. Urban | Rural |
| College name | Wa NTC or Lawra NTC | 0. Lawra NTC 1. Wa NTC | Lawra NTC |
| Programme of study | RGN or NA | 0. NAC 1. RGN | NAC |
| Level of study | First or second year | 0. First year 1. Second year | First year |
| Parent/guardian's level of education | Below tertiary or tertiary education | 0. Below tertiary 1. Tertiary | Below tertiary |

NTC- Nursing training college; RGN- Registered general nurse; NAC- Nurse assistant curative

## Ethical considerations

The Kwame Nkrumah University of Science and Technology/ School of Medical Sciences' Committee on Human Research Publications and Ethics (CHRPE/AP/392/20) granted the researchers ethical clearance to conduct the survey. Institutional access was granted by the Upper West Regional Health Director and the principals of the Wa and Lawra NTCs. Written informed consent was obtained from all the study participants before the conduct of the study. The survey was carried out in accordance with the principles of the Declaration of Helsinki [33].

## Results

### Socio-demographic characteristics of the respondents

Table 3 shows the respondents' demographic characteristics. The survey was completed by 402 of the 410 nursing students sampled, yielding a response rate of 98.2%. Non-responses were due to a lack of interest in the study. The majority of the respondents (71.4%) were female. Over half (54.5%) of them were over the age of 23. The majority of nursing students (90.8%) had never married. 61.9% were Christians. The Dagaaba/Waala/Sisaala, Akan, and other ethnic groups were represented by 55.5, 24.4, and 20.1% of respondents, respectively. Urban

**Table 3. Socio-demographic characteristics of the nursing students (n = 402).**

| Variables | Frequency | Percentage (%) | HBsAb titre testing | | χ2 (df), p-value |
|---|---|---|---|---|---|
| | | | No | Yes | |
| **Sex** | | | | | 0.90 (1), *p* = 0.345 |
| Female | 287 | 71.4 | 234 | 53 | |
| Male | 115 | 28.6 | 89 | 26 | |
| **Age** | | | | | 1.56 (1), *p* = 0.211 |
| < 23 years | 183 | 45.5 | 152 | 31 | |
| ≥ 23 years | 219 | 54.5 | 171 | 48 | |
| **Marital status** | | | | | 4.2 (1), *p* = 0.040 |
| Married | 37 | 9.2 | 25 | 12 | |
| Never married | 365 | 90.8 | 298 | 67 | |
| **Religion** | | | | | 3.3 (1), *p* = 0.068 |
| Islam | 153 | 38.1 | 130 | 23 | |
| Christianity | 249 | 61.9 | 193 | 56 | |
| **Ethnicity** | | | | | 0.93 (1), *p* = 0.334 |
| Dagaaba/Waala/Sisaala | 223 | 55.5 | 183 | 40 | |
| Others | 179 | 44.5 | 140 | 39 | |
| **Residential setting** | | | | | 1.5 (1), *p* = 0.221 |
| Rural | 194 | 48.3 | 151 | 43 | |
| Urban | 208 | 51.7 | 172 | 36 | |
| **Name of College** | | | | | 9.2 (1), *p* = 0.002 |
| Lawra NTC | 177 | 44.0 | 131 | 47 | |
| Wa NTC | 225 | 56.0 | 192 | 32 | |
| **Programme of Study** | | | | | 0.4 (1), *p* = 0.530 |
| NAC | 201 | 50.0 | 159 | 42 | |
| RGN | 201 | 50.0 | 164 | 37 | |
| **Level of study** | | | | | 0.2 (1), *p* = 0.657 |
| First year | 287 | 71.4 | 229 | 58 | |
| Second year | 115 | 28.6 | 94 | 21 | |
| **Parents' level of education** | | | | | 15.4 (1), *p* < 0.0001 |
| Below Tertiary | 236 | 58.7 | 205 | 31 | |
| Tertiary | 166 | 41.3 | 118 | 48 | |

NTC- Nursing training college; RGN- Registered general nurse; NAC- Nurse assistant curative

nursing students (Wa College) made up 51.7% of the respondents. The majority (56.0%) were Wa nursing students. The RGN and NAC programmes had equal representation. The majority (71.0%) of the respondents were first-year nursing students. The majority (58.7%) of students had parents with less than tertiary education.

## Hepatitis B testing, hepatitis B vaccination, number of vaccine doses and anti-HBs testing

The frequencies of hepatitis B testing, vaccination, and post-vaccination anti-HBs titre testing among the nursing students are shown in Table 4. Most of the nursing students (89.0%) had been tested for hepatitis B infection. The majority (72.1%) had received at least one dose of the hepatitis B vaccine. Only 59.5% of the respondents had received three or four doses of the hepatitis B vaccine. Only 19.4% of the respondents had undergone post-vaccination anti-HBs titre testing.

**Table 4. Hepatitis B testing, vaccination, vaccine doses and anti-HBs testing (n = 402).**

| Variable | Category | Frequency (%) |
|---|---|---|
| Hepatitis B testing | No | 44 (11.0) |
| | Yes | 358 (89.0) |
| Hepatitis B vaccination | No | 112 (27.9) |
| | Yes | 290 (72.1) |
| Number of doses | Less than three | 163 (40.5) |
| | Three/Four | 239 (59.5) |
| Anti-HBs titre testing | No | 324 (80.6) |
| | Yes | 78 (19.4) |

## Predictors of anti-HBs titre testing

The results of the predictors of anti-HBs titre testing have been presented in Table 5. As shown in Table 5, hepatitis B vaccination ($p < 0.05$), receipt of $\geq 3$ doses of vaccine ($p < 0.001$), being a student of Wa NTC ($p < 0.001$), and parental tertiary education status ($p < 0.01$) were statistically significantly associated with post-vaccination anti-HBs titre testing.

Nursing students who were vaccinated compared to the unvaccinated were more than 12 times more likely [aOR = 12.34; 95% CI = 1.80–84.54] to access anti-HBs titre testing. Nursing students who had received $\geq 3$ doses of the hepatitis B vaccine had more than eight times the odds [aOR = 8.31; 95% CI = 2.73–25.34] of utilising anti-HBs titre testing services. Students of Wa NTC, compared to Lawra nursing students, were 74% less likely [aOR = 0.26; 95% CI = 0.15–0.47] to access anti-HBs titre testing. Finally, when compared to nursing students whose parents had less than tertiary education, nursing students whose parents had attained tertiary education status were more than two times more likely [aOR = 2.50; 95% CI = 1.42–4.42] to access anti-HBs titre testing.

## Discussion

This study assessed the prevalence of hepatitis B testing, hepatitis B vaccination, the number of vaccine doses, anti-HBs titre testing, and the predictors of anti-HBs titre testing among nursing students in Northwest Ghana.

The study revealed a high hepatitis B testing rate among the nursing students. The high hepatitis B testing rate reflects proactive health-seeking behaviour and a demonstration of awareness of risks among the nursing students. Similar findings have been reported by Ma et al. [34] among Vietnamese Americans. However, Osei et al. [4] in Ghana and Kue and Thorburn [35] in the USA reported lower hepatitis B testing rates. These differences in findings can be explained in terms of relative availability and access to testing services. High hepatitis B testing rates imply that those students who test negative for HBsAg and have not been vaccinated can benefit from the hepatitis B vaccine while infected people receive treatment [36]. Based on this finding, management of health training institutions should enforce and provide subsidised hepatitis B testing services to ensure universal testing of nursing students.

This study reported low complete hepatitis B vaccination and post-vaccination anti-HBs titre testing rates among the nursing students. This implies that the high hepatitis B testing rate among the nursing students did not translate into increased rates of completing the vaccine series and subsequently accepting the post-vaccination anti-HBs titre testing. This finding also reflects the dearth of proactiveness of the nursing students in seeking healthcare services. This finding corresponds with the reports of Paul and Peterside [36] in Nigeria, Ayalew and Horsa [37] in Ethiopia, and Yuan et al. [12] in China, who also reported low hepatitis B

**Table 5. Predictors of anti-HBs titre testing (n = 402).**

| Variables | Model 0 | Model 1 | Model 2 | Model 3 |
|---|---|---|---|---|
| | aOR [95% CI] | aOR [95% CI] | aOR [95% CI] | aOR [95% CI] |
| **Fixed effects results** | | | | |
| **Hepatitis B testing** | | | | |
| No | | Ref | | Ref |
| Yes | | 0.33 [0.08–1.31] | | 0.31 [0.07–1.31] |
| **Hepatitis B vaccination** | | | | |
| No | | Ref | | Ref |
| Yes | | 10.03* [1.43–70.18] | | 12.34* [1.80–84.54] |
| **Number of vaccine doses** | | | | |
| < 3 doses | | Ref | | Ref |
| ≥ 3 doses | | 6.31** [2.16–18.45] | | 8.31*** [2.73–25.34] |
| **Marital status** | | | | |
| Married | | | Ref | Ref |
| Never married | | | 0.45* [0.21–0.96] | 0.54 [0.23–1.25] |
| **College** | | | | |
| Lawra NTC | | | Ref | Ref |
| Wa NTC | | | 0.47** [0.28–0.78] | 0.26*** [0.15–0.47] |
| **Parent's education** | | | | |
| Below Tertiary | | | Ref | Ref |
| Tertiary | | | 2.40** [1.43–3.96] | 2.50** [1.42–4.42] |
| **Random effects results** | | | | |
| **Constant** | Ref | 0.02*** [0.00–0.12] | 0.49 [0.21–1.16] | 0.03** [0.00–0.25] |
| **Observations** | 402 | 402 | 402 | 402 |
| **Wald $\chi 2$** | Ref | $\chi 2$ (3) = 25.76 | $\chi 2$ (3) = 24.16 | $\chi 2$ (6) = 53.72 |
| **Prob > chi$^2$** | Ref | < 0.0001 | < 0.0001 | < 0.0001 |
| **Penalized log likelihood** | -190.522 | -161.9105 | -180.5224 | -139.6958 |
| **McFadden R$^2$** | Ref | 0.166 | 0.067 | 0.267 |
| **AIC** | Ref | 331.821 | 369.045 | 293.392 |
| **BIC** | Ref | 347.807 | 385.031 | 321.367 |

aOR = Adjusted Odds Ratio; CI = Confidence interval; AIC = Akaike's information criterion; BIC = Bayesian information criterion

*ρ < 0.05

**ρ < 0.01

*** ρ < 0.001

vaccination rates among medical students and healthcare workers. In contrast to our finding, Muvunyi et al. [38] in Rwanda reported a high complete hepatitis B vaccination rate among healthcare workers. However, Park et al. [39] and Abiola et al. [15] reported lower complete hepatitis B vaccination rates. Reports on post-vaccination anti-HBs titre testing rates are extremely rare. In a study conducted by Aroke et al. [5], none of the respondents had done the post-vaccination anti-HBs titre testing. However, Di Giampaolo et al. [40] reported a higher post-vaccination anti-HBs testing rate among healthcare trainees in Central Italy. These differences are related to their level of knowledge, relative availability, and accessibility to hepatitis B vaccination and post-vaccination anti-HBs titre testing services [19, 20, 41]. Studies conducted in countries where hepatitis B vaccination and post-vaccination anti-HBs titre testing services are readily available and accessible at a subsidised cost and at near-to-reach centres tend to report higher full hepatitis B vaccination and post-vaccination anti-HBs titre testing rates [38, 40]. These findings can also be explained by Andersen's healthcare utilisation model and the

healthcare access dimensions [19, 20]. Differences in predisposing factors (socio-demographics like age, sex, marital status, residential setting), enabling factors (socioeconomic status, health insurance), need factors (need awareness, perceived need), as well as relative availability, geographic accessibility, and financial affordability may be the reasons for the reported low rates of hepatitis B vaccination and post-vaccination anti-HBs titre testing among the nursing students [19, 20]. There are no designated hepatitis B vaccination centres in the UWR, and when available, they are usually not in close proximity to these nursing students. Post-vaccination anti-HBs titre testing services in particular are unavailable in the UWR. Moreover, transported samples for post-vaccination anti-HBs titre testing poses a financial access barrier [42, 43]. Low rates of complete hepatitis B vaccination and post-vaccination anti-HBs titre testing are detrimental to the object of hepatitis B infection prevention and control. This study, therefore, recommend the provision of affordable vaccination and post-vaccination anti-HBs titre testing facilities that are in close proximity to these training institutions.

However, students of Wa NTC were surprisingly statistically significantly less likely to utilise the post-vaccination anti-HBs titre testing services. Wa NTC is located in a predominantly urban locality and, therefore, is expected to be inequitably more advantaged in respect of availability and geographical accessibility to testing centres [20]. However, considering that post-vaccination anti-HBs titre testing is generally unavailable in the UWR, these factors probably become generally less important explainable factors. Additionally, rural localities are relatively advantaged in terms of outreach services by non-profit organizations because of their pro-poor nature. Therefore, the rural based Lawra NTC may have benefited from financial assistance from these organizations towards utilization of post-vaccination anti-HBs titer testing.

However, further research is required to explore other factors that favour rural localities with respect to post-vaccination anti-HBs titre testing.

Furthermore, nursing students whose parents or guardians have had tertiary education were statistically significantly more likely to access post-vaccination anti-HBs titre testing. This finding can be interpreted that, there is an influence of parental education on the utilisation of anti-HBs titre testing services. This finding is in similitude with the findings of Mora and Trapero [17] in Spain among children enrolled in a vaccination programme. Highly educated parents, in line with Andersen's healthcare utilisation model, are probably more informed in respect of access to health information and are therefore more likely to ensure their wards access post-vaccination anti-HBs titre testing [19]. This is particularly important because of the generally low awareness associated with post-vaccination anti-HBs titre testing [5]. Additionally, higher education improves socio-economic status [44]. Therefore, highly educated parents who are socio-economically more advantaged compared to parents who have a lower tertiary education status are plausibly more likely to be able to break the financial affordability barrier associated with access to post-vaccination anti-HBs titre testing [19]. This finding calls for tailored health education interventions aimed at improving access to and utilisation of post-vaccination anti-HBs titre testing, particularly among nursing students. Management of nursing training colleges should therefore liaise with the Ministry of Health, Ghana Health Service, and healthcare delivery agencies to implement tailored health education and the provision of affordable hepatitis B testing, vaccination, and post-vaccination anti-HBs titre testing services.

The study also revealed no statistically significant association between hepatitis B testing and anti-HBs titre testing. This may be due to the mandatory health testing requirement before enrolment in nursing colleges visa viz a progressive increase in the awareness level of these nursing students in relation to the role of testing in the diagnosis of hepatitis B infection [45]. Although marital status was a significant predictor of utilisation of anti-HBs titre testing services among the nursing students, it showed no significance in the final model (model 3). However, Paul and Peterside [46] in Nigeria reported otherwise among nursing and medical

students. Variations in levels of experience, awareness, and exposure to education could be the reason for the reported differences in findings.

### Strengths and limitations of the study

This study is pioneering, being the first to explore the relationship between utilisation of anti-HBs titre testing and hepatitis B testing, hepatitis B vaccination, number of vaccine doses, and socio-demographics among nursing students in Ghana. A reliable questionnaire with good internal consistency was administered. The inclusion of follow-up calls enhanced internal validity of the results, as it counteracted the low response rate associated with online surveys. Achieving a 98.2% response rate, therefore, bolstered the external validity of the study. To minimise selection bias and enhance representativeness, probabilistic multi-stage sampling was employed. Rigorous hierarchical data analysis was conducted to eliminate spurious factors. However, the representativeness of the study may be limited since only two out of seven nursing schools in Northwest Ghana were sampled, and some social desirability bias could arise due to self-reported results.

### Conclusion

The study reported a high hepatitis B testing rate but lower rates of hepatitis B vaccination, receipt of adequate vaccine doses, and post-vaccination anti-HBs titre testing. Hepatitis B vaccination, receipt of $\geq 3$ vaccine doses, being a student of Wa NTC, and parental tertiary education were independent predictors of anti-HBs titre testing. This study proposes that hepatitis B testing, hepatitis B vaccination, and post-vaccination anti-HBs titre testing should be a non-negotiable pre-requisite for nursing school admission. Additionally, easily accessible and affordable hepatitis B testing, vaccination, and post-vaccination anti-HBs titre testing services covered by the National Health Insurance Scheme should be provided. These interventions should focus on unvaccinated nursing students, students who have had $< 3$ vaccine doses, Wa NTC students, and students whose parental education is below tertiary status.

### Supporting information

**S1 Data. Minimum dataset.**
(XLSX)

### Author Contributions

**Conceptualization:** Augustine Ngmenemandel Balegha.

**Data curation:** Augustine Ngmenemandel Balegha.

**Formal analysis:** Augustine Ngmenemandel Balegha.

**Investigation:** Augustine Ngmenemandel Balegha.

**Methodology:** Augustine Ngmenemandel Balegha.

**Project administration:** Augustine Ngmenemandel Balegha.

**Resources:** Augustine Ngmenemandel Balegha.

**Supervision:** Augustine Ngmenemandel Balegha.

**Writing – original draft:** Augustine Ngmenemandel Balegha.

**Writing – review & editing:** Augustine Ngmenemandel Balegha.

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
