## [Decision Letter · Decision Letter 0]

5 Mar 2024

PGPH-D-23-01827

Factors influencing the utilisation of anti-HBs titre testing services among nursing students in Northwest Ghana: A cross-sectional study

Dear Dr. Balegha,

Thank you for submitting your manuscript to PLOS Global Public Health. After careful consideration, we feel that it has merit but does not fully meet PLOS Global Public Health’s publication criteria as it currently stands. Therefore, we invite you to submit a revised version of the manuscript that addresses the points raised during the review process.

We look forward to receiving your revised manuscript.

Kind regards,

Javier H Eslava-Schmalbach, M.D., Ph.D., MSc

Academic Editor

Journal Requirements:

Additional Editor Comments (if provided):

Dear Authors:

We are pleased to inform you that we have finally received the reviewers' comments on your manuscript.

Also to the Reviewers' comments, you should consider these Editor's comments:

Table 1: Discrepancies exist between the total population per class and the sum of individual rows. Please double-check all numerical values throughout the manuscript.

Tables 4 & 5: The analysis included fewer respondents than initially anticipated. Address the potential impact of this data loss on the precision of your results. Refer to these guidelines for more information: https://www150.statcan.gc.ca/n1/en/catalogue/12-539-X (pp. 1-49).

Table 5: The combination of high OR values and wide confidence intervals suggests that logistic regression might not be the most suitable model for this analysis. The authors mention that logistic regression can overestimate results with low frequencies, as noted in https://pubmed.ncbi.nlm.nih.gov/20213709/. In such cases, it's recommended to consider Poisson or binomial regression models instead.

Please address each of these points and the reviewers' comments in your response. We look forward to receiving your revisions as soon as possible.

Sincerely,

Javier Eslava-Schmalbach

Reviewers' comments:

Reviewer's Responses to Questions

**Comments to the Author**

1. Does this manuscript meet PLOS Global Public Health’s publication criteria? Is the manuscript technically sound, and do the data support the conclusions? The manuscript must describe methodologically and ethically rigorous research with conclusions that are appropriately drawn based on the data presented.

Reviewer #1: Yes

Reviewer #2: Partly

2. Has the statistical analysis been performed appropriately and rigorously?

Reviewer #1: I don't know

Reviewer #2: Yes

3. Have the authors made all data underlying the findings in their manuscript fully available (please refer to the Data Availability Statement at the start of the manuscript PDF file)?

Reviewer #1: Yes

Reviewer #2: Yes

4. Is the manuscript presented in an intelligible fashion and written in standard English?

Reviewer #1: Yes

Reviewer #2: Yes

5. Review Comments to the Author

Reviewer #1: While an effort has been made in the research, with a good sample size and laboratory tests, I consider there is no significant contribution to the topic. It is not specified which laboratory test and technique were used. It's unclear whether the titers of anti-HBsAg were determined or if the test was only qualitative. I believe the study's objective should be clarified, for example, whether there is a relationship between the titers of anti-HBsAg antibodies and the applied doses, age, gender, etc

Reviewer #2: Your study, "Factors influencing the utilisation of anti-HBs titre testing services among nursing students in Northwest Ghana: A cross-sectional study," is well written and provides valuable insights into the hepatitis B vaccination and testing behaviors among nursing students in a specific region of Ghana.

Here are some comments and suggestions for improvement:

Abstract

The structured abstract provides a comprehensive and coherent overview of the study, from the background through to its conclusions. It effectively sets the stage for the full article by highlighting the significance of the research, its methodology, key findings, and implications. However, specifically after the introduction in the abstract, it would be beneficial to clearly articulate the objectives or aim of the study

Introduction

While the introduction effectively outlines the significance of hepatitis B prevention and the need for research among nursing students in Northwest Ghana, it lacks explicit statements of research objectives.

Methods

1. For enhancing the geographical specificity and contextual understanding of your study's settings, it's recommended to include the latitude and longitude coordinates of the Wa and Lawra Nursing Training Colleges (NTCs).

2. Clarifying the application of Cochran's formula for sample size calculation without accounting for the finite population of nursing students at Wa and Lawra NTCs is crucial, as adjustments or a correction factor may be needed to prevent overestimation. An explanation of whether a finite population correction was used and its impact on the sample size calculation would significantly enhance the methodological rigor.

3. Additionally, justifying the decision to increase the sample size by 20% to account for non-responses would improve the study's methodological clarity. It's important to detail the basis for this adjustment, whether it stems from prior studies, pilot data, standard practices, or anticipated challenges specific to this research context.

Ethical clearance

Could the author clarify whether, in addition to the granted institutional access by the Upper West Regional Health Director and the principals of the Wa and Lawra NTCs, there was a requirement or consideration for obtaining additional institutional ethical clearance, specifically from the institutions where the research was conducted?

Discussion

1. Lines 255 to 2587….Could the author consider comparing the high hepatitis B testing rate found among the nursing students in this study with similar populations of nursing students, if such data are available? This comparison could provide a more nuanced understanding of the health-seeking behaviors and awareness levels specific to nursing students, potentially highlighting unique factors influencing these rates in different educational or cultural settings.

2. Line 271 in what kind of population pls?

3. Lines 297 to 300: The finding that Wa NTC students in an urban area were significantly less likely to utilize post-vaccination anti-HBs titre testing services contradicts the typical assumption that urban settings, with their better access to healthcare facilities, would lead to higher service utilization. Could the author revisit this discussion to explore potential reasons or factors behind this unexpected pattern, without necessarily altering the study's findings?

4. Line 307 pls specify the pop.

5. Lines 331 to 333 the first in Ghana or globally?

6. PLOS authors have the option to publish the peer review history of their article (what does this mean?). If published, this will include your full peer review and any attached files.

**Do you want your identity to be public for this peer review?** For information about this choice, including consent withdrawal, please see our Privacy Policy.

Reviewer #1: No

Reviewer #2: **Yes: **Bashar Haruna Gulumbe

---

## [Editor Report · Decision Letter 1]

5 Apr 2024

Factors influencing the utilisation of anti-HBs titre testing services among nursing students in Northwest Ghana: A cross-sectional study

PGPH-D-23-01827R1

Dear Dr. Balegha,

We are pleased to inform you that your manuscript 'Factors influencing the utilisation of anti-HBs titre testing services among nursing students in Northwest Ghana: A cross-sectional study' has been provisionally accepted for publication in PLOS Global Public Health.

Best regards,

Javier H Eslava-Schmalbach, M.D., Ph.D., MSc

Academic Editor